# Design and Testing of Brushless DC Motor Components of A6 Steel Additively Manufactured by Selective Laser Sintering

Sebastian-Marian Zaharia [1,*], Mihai Alin Pop [2], George Razvan Buican [1], Lucia-Antoneta Chicos [1], Valentin Marian Stamate [1], Ionut Stelian Pascariu [1] and Camil Lancea [1]

[1]   Department of Manufacturing Engineering, Transilvania University of Brasov, 500036 Brasov, Romania
[2]   Department of Materials Science, Transilvania University of Brasov, 500036 Brasov, Romania
*    Correspondence: zaharia_sebastian@unitbv.ro

**Abstract:** Metallic additive manufacturing technology is seeing increasing use from aviation companies manufacturing prototypes or components with complex geometric shapes, which are then tested and put into operation. This paper presents the design, fabrication via a selective laser sintering process, and testing of the mechanical performance by performing three-point bending and tensile tests on A6 steel specimens. After performing the mechanical tests on specimens made from A6 steel manufactured via the SLS process, the following performances were obtained: the maximum three-point bending strength was 983.6 MPa and the maximum tensile strength was 398.6 MPa. In the microscopic analysis of the specimens manufactured by the selective laser sintering process, a homogeneous structure with defects specific to additive processes (voids) was revealed. Additionally, the feasibility of designing, manufacturing through the selective laser sintering process and subsequent testing of some components (rotor, right case, left case and motor mount) from a brushless motor made from A6 steel material was demonstrated. After testing the brushless motor, the main performances showed stable behavior of the motor and a linear dependence with the increase in electronic speed control signal or motor electrical speed, resulting in a maximum thrust force of 4.68 kgf at 7800 RPM.

**Keywords:** brushless motor; performance analysis; selective laser sintering; mechanical testing

## 1. Introduction

A brushless direct current (BLDC) motor is a synchronous motor that uses a direct current (DC) power supply. The BLDC motor uses an electronic closed-loop controller to switch DC currents to the motor windings, producing magnetic fields that actually rotate in space and which the permanent magnet rotor follows [1]. The controller adjusts the phase and amplitude of the DC pulses to control the speed and torque of the motor [2]. The main advantages of a brushless motor are long life, high power-to-weight ratio, high speed and efficiency, low manufacturing and maintenance costs, almost instantaneous speed and torque control and its compact structure [3]. In recent years, BLDC motors have seen frequent use in fields such as aerospace, automotive, robotics, household appliances, food and chemicals, medical equipment and computer peripherals [4,5]. Information on the performance of BLDC motors used in aviation to propel unmanned aerial vehicles (UAVs) is generally available from the suppliers of such propulsion systems. However, an accurate estimate of the performance of BLDC motors depends on the materials from which the stator and rotor cores are made, their geometry, the type and number of permanent magnets and the winding distribution. In order to determine the performance of BLDC motors, various studies were carried out using various techniques and methods. The behavior of BLDC motors was analyzed using two methods: Fast-Fourier Transform (FFT) and chaos analysis [6], and application of the cuckoo search (CS) algorithm in an attempt to minimize the commutation torque ripple in a brushless BLDC motor [7], improving the

efficiency and reducing the acoustic noise of BLDC motors by reducing current harmonics using a novel commutation method to generate a sinusoidal current waveform without requiring additional hardware [8]. Another study [9] investigates three BLDC motor control schemes analyzed under different operating conditions, including steady-state and transient operations. A comparative study based on qualitative, quantitative and electromagnetic analysis of BLDC motors was carried out using finite element analysis in order to optimize the designed models [10,11]. BLDC motors are intensively studied and used in various aerospace applications, such as analysis of the flight performance of drones [12–14] and UAVs [15,16], as well as various morphing wing models, actuated by actuator systems, increasing their span/chord during flight [17,18].

Currently, additive manufacturing technologies are successfully used in various fields, such as aviation, automotive, medicine, dentistry, architecture, various aircraft [19–21], functional components [22–24], models for assembly testing [25] and medical device prototypes [26–28]. Additive technology has represented and continues to represent a challenge, especially in the field of aviation where there is fierce competition in the market to produce aeronautical products, in the shortest possible time, that respect both aeronautical regulations and structural performances. Additive manufacturing technologies represent a current trend in the realization of electric machines or their components, offering a high degree of flexibility regarding the design of products in order to achieve performance parameters which are more difficult to achieve through conventional manufacturing techniques [29,30]. Manufacturing a BLDC motor, completely through additive manufacturing technologies remains a challenge for manufacturing and aviation engineers. Next, some studies on certain components of electric motors made using additive manufacturing technologies will be presented. In order to verify the design process, an initial study [31] was carried out regarding selective laser melting (SLM) manufacturing of three components from a synchronous reluctance motor (SynRM). One option for making BLDC motors is manufacturing through the fused filament fabrication (FFF) process. In this sense, an electric motor with permanent magnets was manufactured and tested using the FFF process from polylactic acid filament [32]. Of course, the design for additive manufacturing is also an important aspect that must be considered. In this sense, various models of spiderweb lattice structures used in a BLDC motor were designed and analyzed using the finite element method [33]. Garibaldi et al. [34] reported the successful manufacture, through the laser beam machining (LBM) process, of the rotor core made of FeSi6.9, for rotating electrical machinery. A recent study [35] demonstrated the feasibility of manufacturing concentrated windings through the laser powder bed fusion (LPBF) process of pure copper, with air between the flat layered turns. Another study [36] showed that that topology optimized soft magnetic core was successfully created using the selective laser melting process, from Fe-Co powder. Lancea et al. [37] manufactured the components (the rotor and the two housings right case and left case) via the FFF process, from LW-PLA filament, with the aim of verifying the assembly of the components and finding design problems.

Thus, this study is based on previous research [37], which expanded the manufacturing domain for a functional model of a BLDC motor made using the selective laser sintering (SLS) process. Starting from the initial model, by modifying and manufacturing the components (rotor, right case, left case and motor mount) using the SLS method, a functional BLDC motor was assembled and tested. Prior to the SLS manufacture of the BLDC motor components mentioned above, in this study, the A6 steel was tested (three-point bending and tensile) and microscopically analyzed.

## 2. Materials and Methods

### 2.1. Design of Specimens Used in Mechanical Tests

By using current standards ASTM E290-14 [38] and ASTM E8/E8M-16ae1 [39] applied to metal specimens manufactured via additive processes, and by using the SolidWorks 2021 software system (Dassault Systèmes SolidWorks Corporation, Waltham, MA, USA), the specific specimens for mechanical tests (three-point bending and tensile) were designed.

The specimens, manufactured by the SLS process, tested in three-point bending, have the dimensions shown in Table 1.

**Table 1.** The dimensions of specimens subjected to three-point bending.

| Length L [mm] | Width w [mm] | Thickness t [mm] |
|:---:|:---:|:---:|
| 130 | 19 | 3.2 |

The specimen dimensions used in the three-point bending and tensile tests were detailed in Table 2.

**Table 2.** The dimensions of specimens subjected to tensile testing.

| Overall Length [mm] | Distance between Grips [mm] | Gauge Length [mm] | Width of Grip Section [mm] | Width [mm] | Thickness [mm] | Radius of Fillet [mm] |
|:---:|:---:|:---:|:---:|:---:|:---:|:---:|
| 165 | 115 | 57 | 19 | 13 | 3.2 | 76 |

### 2.2. Component Design for BLDC Motor Model

The design of the BLDC motor components from the previous study [32] was optimized for a more appropriate additive manufacturing process. Thus, for easier manufacturing, the rotor and the right housing were designed as one piece (Figure 1a), to remove the need for assembly. This component (Figure 1a) has been provided with spacer surfaces for easy positioning of the magnets. The other two designed components of the BLDC motor were the left housing (Figure 1b) and the motor mount (Figure 1c). The rotor has a wall thickness of 2 mm, a length of 55 mm and an outer diameter of 49.8 mm. The left case (Figure 1b) has a diameter of 49.8 mm, a shell thickness of 2 mm and is provided with holes for ventilation. The motor mount component (Figure 1c) has four arms and a thickness of 4 mm.

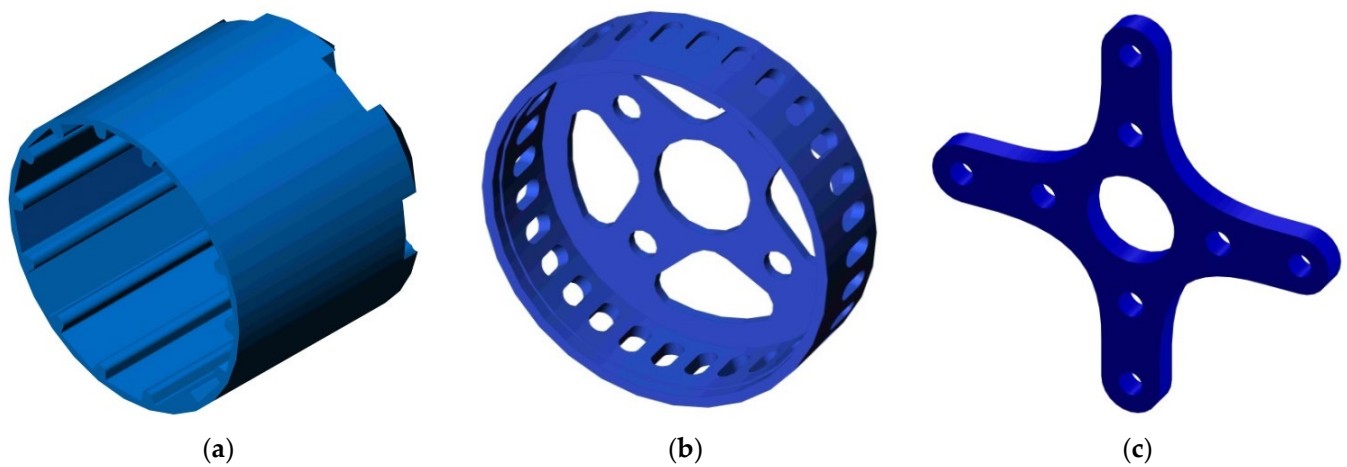

(**a**)                                                       (**b**)                                                       (**c**)

**Figure 1.** Designed components of BLDC motor: (**a**) rotor; (**b**) left case; (**c**) motor mount.

### 2.3. SLS Manufacturing of BLDC Motor Specimens and Components

After careful analysis of the design and operation of the electric motor, the following components (rotor and right housing, printed together, left housing and flange) are good candidates for additive manufacturing using the SLS process. The other components of the motor (stator with winding, shaft) were manufactured using classical manufacturing technologies. The 3D System sPRO 60 SD printer (SPRO 60, 3D Systems, Inc., Rock Hill, SC, USA) was used to manufacture the BLDC motor components. The 3D printing facility allows the creation of highly complex products, from a geometrical point of view, using

metal, ceramic, polymer and composite powders. The sintering furnace attached to the SLS printing plant ensures the sintering of ceramic and metal powders and infiltration with different metals.

If, in the case of using polymeric materials, the final component is obtained directly from the SLS process, without the need for other subsequent steps, in the case of using metal powder, after obtaining the desired components, an additional bronze infiltration step is required. This additional step is preceded by a weighing of the component and the establishment of the infiltrant proportion (65% of the mass of the component), according to the technical data sheet provided by the manufacturer. These bronze particles are placed in contact with the specimens to guarantee infiltration via capillary action once melting takes place in the furnace. Components and bronze are coated with alumina powder for homogeneous heat transfer and infiltration. Infiltration is a post-processing method [40] of components manufactured by the SLS process from metal powder. Infiltration is used in particular to densify and strengthen components when the required characteristics cannot be achieved through the main manufacturing process (SLS in this study).

The material used in the manufacture of the electric motor components, through the SLS process, was A6 steel, produced by the 3D Systems Company. LaserForm™ A6 steel powder is a magnetic material, used for rapid prototyping of molds and tools, which presents the following advantages: design flexibility, excellent machinability, good thermal conductivity and surface finish [41].

BLDC motor specimens and components, designed in SolidWorks 2021, were exported in .stl (stereolithography) format for manufacture using the 3D System sPRO 60 SD printer. The Vanguard software system was used to generate the machine code for the SLS manufacturing process of the components. In Table 3, the main manufacturing parameters used in the SLS process to make the BLDC motor components and the specimens are presented.

**Table 3.** Manufacturing parameters of BLDC motor components and specimens.

| Process Parameter | Unit | Value |
|---|---|---|
| Fill laser power | W | 15 |
| Outline laser power | W | 5 |
| Layer thickness | μm | 100 |
| Scanning speed | m/s | 5 |
| Laser beam diameter | μm | 60 |
| Preheating temperature | °C | 120 |

After manufacturing the components of the BLDC motor (Figure 2a–c), the next steps were to clean and weigh the parts, calculate the amount of infiltrate (Figure 2d) and place them in the heat-treatment furnace. The infiltration temperature was 1070 °C, according to the plant manufacturer's specifications, in gaseous nitrogen atmosphere, and the total heating–holding–cooling cycle was 72 h. The last stage was the removal of the "tabs" (supports created intentionally for placing the bronze) and the turning process on a lathe. The results of the SLS additive process can be seen in Figure 2e (BLDC motor rotor) and in Figure 2f (specimens used in mechanical tests).

*2.4. Test Conditions and Microscopic Analysis of Specimens*

Three-point bending testing (Figure 3a) and tensile testing (Figure 3b) of the specimens, manufactured by the SLS process, was performed on a WDW-150S universal testing machine (Jinan Testing Equipment IE Corporation, Jinan, China). For testing, five specimens were manufactured for each type of test, according to standards for the testing of mechanical specimens. Three-point bending tests and tensile tests were carried out in order to determine the characteristics (bending strength, tensile strength and aspects related to the load—displacement characteristic curve) of the A6 steel material. This information was the basis for the use of A6 steel material in the manufacture of BLDC motor components. Both types of tests were carried out with a loading speed of 5 mm/min [42,43], until the

specimens broke. In order to assess the manufacturing method of the A6 steel material, microscopic analyses were performed using three sections (Figure 3c), with the help of a Nikon Eclipse MA 100 microscope (Nikon Corp., Tokyo, Japan). The specimens were polished and treated with aqua regia to reveal the internal structure of the A6 steel material.

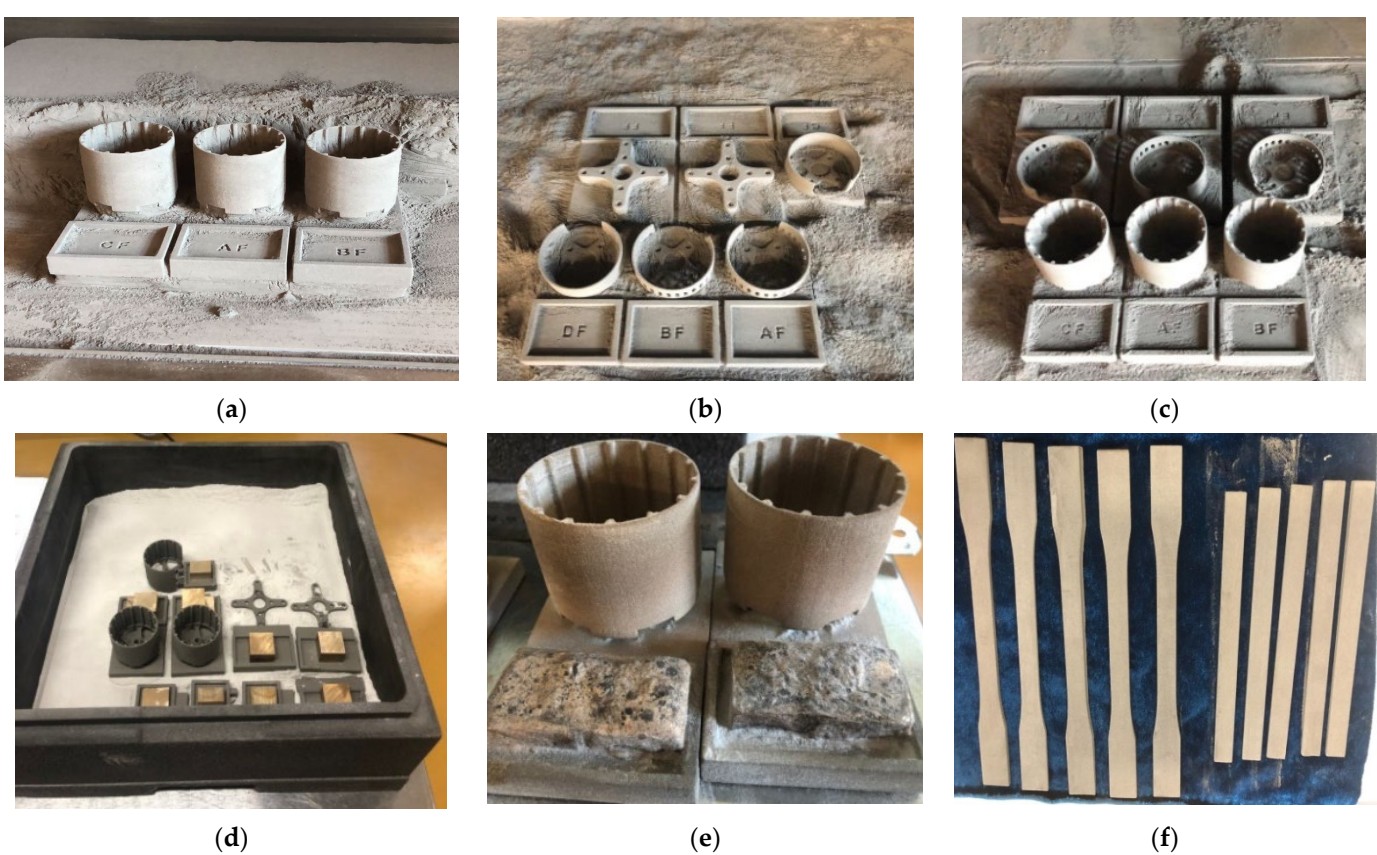

**Figure 2.** Additive manufacturing of BLDC motor components using the SLS process: (**a**) Parts cleaning; (**b**) left housing and motor mount; (**c**) rotor; (**d**) preparing the components for infiltration into the sintering furnace; (**e**) rotor obtained after infiltration; (**f**) specimens obtained after infiltration.

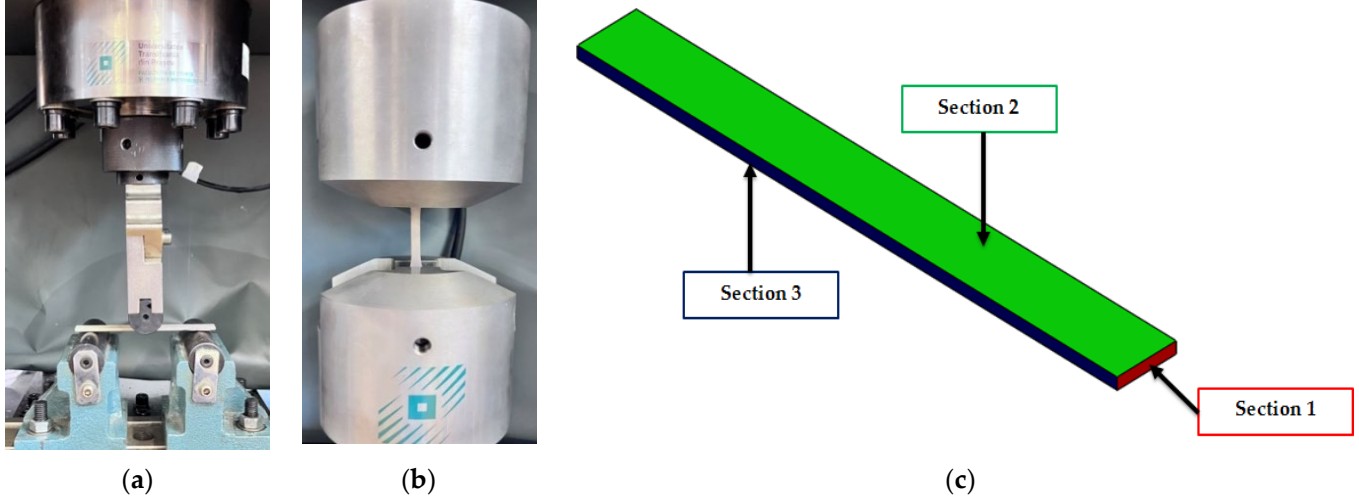

**Figure 3.** Testing and analysis of specimens: (**a**) three-point bending test; (**b**) tensile testing; (**c**) establishment sections for microscopic analysis.

### 2.5. Assembling BLDC Motor Components

In the first stage of the assembly of the BLDC motor manufactured via the SLS process, the centering and fixing of the rotor component on the motor shaft was considered in order to check the position of the shaft and the mode of operation of the rotor. Afterwards, the winding support was fixed on the shaft and two radial ball bearings were inserted in the specially created bearing housing from the right cover. Due to the high complexity and precision of the execution, it was decided that the stator and winding should be manufactured by a specialized manufacturer. This is because the manufacturing of magnets and copper windings by additive technologies has not reached the technological maturity and performance of the same products made using classical technologies.

The stator component contains 12 coils wrapped with 0.25 mm diameter copper wire. The 14 neodymium magnets were glued to the rotor housing with two-component epoxy adhesive using the spacer support (Figure 4a), made via the SLS process. The magnets have been considered to follow the configuration of a brushless electric motor, so the magnets inside the rotor are attached in alternating polarity and therefore attract and repel each other. For the dynamic balancing of the rotor, it required various machining operations: turning and internal/external grinding (Figure 4b).

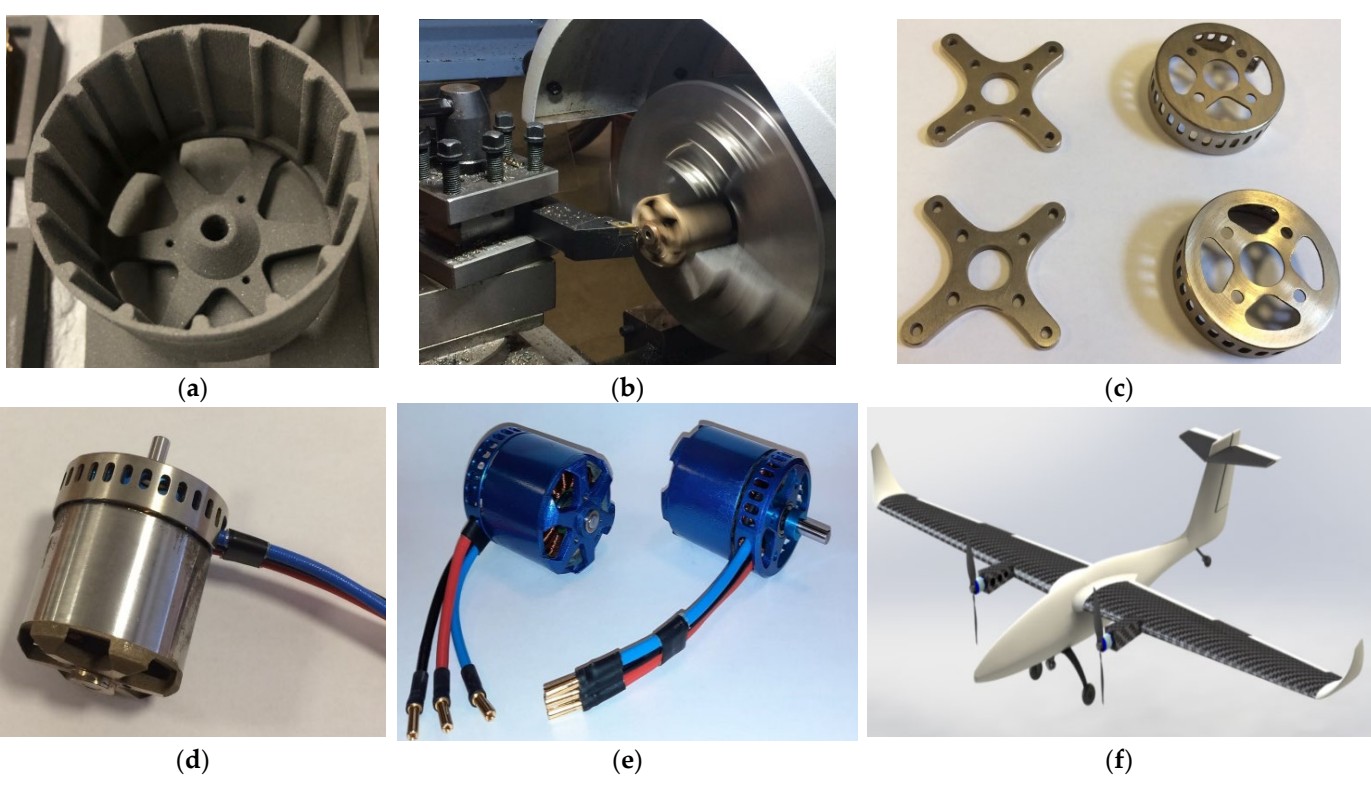

**Figure 4.** Assembling BLDC motor components: (**a**) rotor fitted with spacer; (**b**) rotor turning; (**c**) motor mount and left housing; (**d**) assembled motor; (**e**) electric motor prepared for testing; (**f**) the UAV aircraft on which the BLDC motors will be mounted.

Next came the assembly of the left cover equipped with a slot for removing the three wires. These wires were obtained from the pairing of four coils on the stator. The next step was to introduce the ball bearings on the stator mount, and they had to absorb all the forces from the BLDC motor (vibration, torque). After the ball bearings had been fully positioned, the shaft and rotor were reinstalled. For a more precise fixation of the ball bearings, a safety ring C was positioned on the shaft. This safety ring C and the step washer held the BLDC motor together. Because the BLDC motor operates at high speeds, for high operational safety, the motor shaft was held in place by two M4 grub screws.

On the left cover of the stator, in the four threaded holes, the motor mount (Figure 4c) is secured by screws, which allows the positioning of the electric motor on the aircraft structure. Once the electric motor was assembled (Figure 4d) the painting preparation followed. The outcome of all these steps was an electric motor with components manufactured using the SLS process (Figure 4e). The BLDC motor will be the propulsion system of a twin-motor airplane, from composite materials and composite sandwich structures [44], through the FFF process (Figure 4f). Table 4 shows the main characteristics of the assembled BLDC motor.

**Table 4.** BLDC motor characteristics.

| Component | Unit | Value |
|---|---|---|
| Rotor diameter | mm | 49.8 |
| Stator diameter | mm | 40.5 |
| Number of poles stator | - | 12 |
| Number of magnets rotor | - | 14 |
| Neodymium magnet dimensions | mm | $30 \times 7.5 \times 2.5$ |
| Length motor | mm | 66 |
| Shaft Diameter | mm | 6.2 |
| Weight | g | 417 |

### 2.6. BLDC Motor Performance Testing

To test the performance of the BLDC motor, manufactured via the selective laser sintering process, the RCbenchmark Series 1585 traction stand [45] was used.

This test stand has the following characteristics: thrust: 5 kgf, torque: 2 Nm, voltage: 50 V, current: 55 A. In order to test the electric motor, we first need to calibrate the RCbenchmark Series 1585 experimental stand. Afterwards, the electric motor is fixed on the stand and first tested without the propeller. When the propeller is mounted, the motor is tested again, but first at low speeds (Figure 5a), and then the speed is increased. The following equipment was used to test the motor: RCbenchmark Series 1585 traction test stand, an ESC (Electronic Speed Controller) carrying a current of 60 A, 2 LiPo batteries with three cells connected in series (Figure 5b), a computer, a RCbenchmark GUI 1.2.0 software system and a $16 \times 8$-inch propeller.

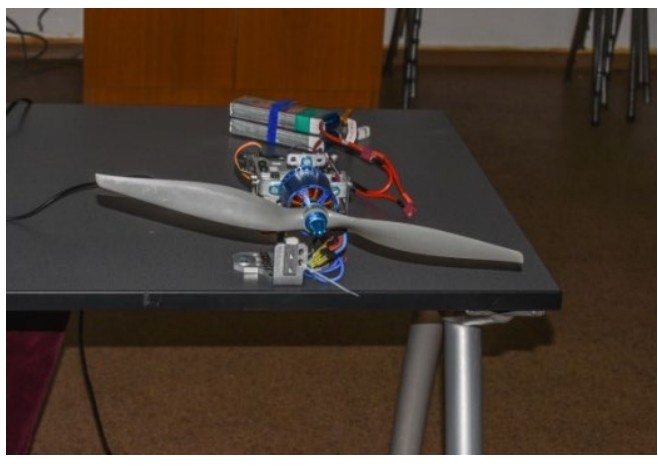

(**a**)

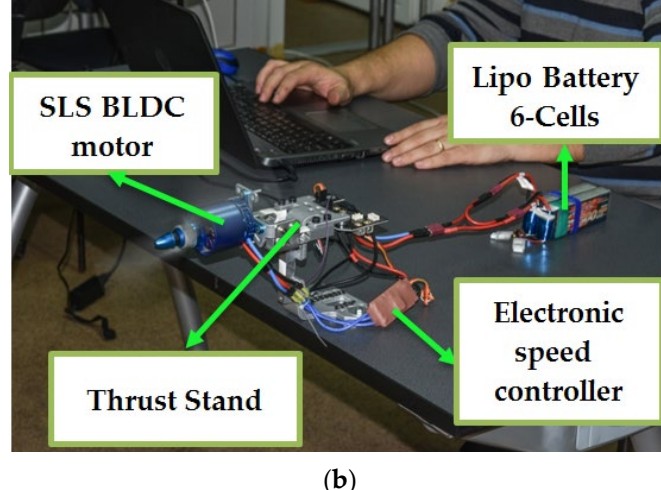

(**b**)

**Figure 5.** BLDC motor testing: (**a**) initial operation check of test stand; (**b**) BLDC motor performance testing.

## 3. Results and Discussion

### 3.1. Three-Point Bending Testing of Specimens Manufactured by the SLS Process

The tests in the static regime, for three-point bending, were carried out on five specimens manufactured by the SLS process from A6 steel, until their failure occurred. The load–displacement behavior for the 5 specimens (Figure 6a), tested in a static bending regime, shows two main stages: a linear increase obtained between the applied force and displacement, with some non-linear behavior, towards the maximum of the curve and then a sudden decrease, at maximum force, at the moment of specimen breakage. It can be seen that the maximum force, until the moment when the irreversible damage appeared in the A6 steel material, was about 0.8 kN. Additionally, the displacement of the specimens manufactured by the SLS process from the A6 steel material, at which the irreversible damage occurred, was 17.5 mm.

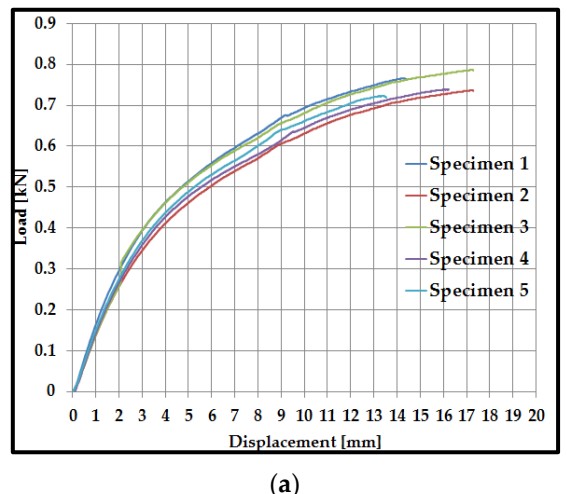

(**a**)

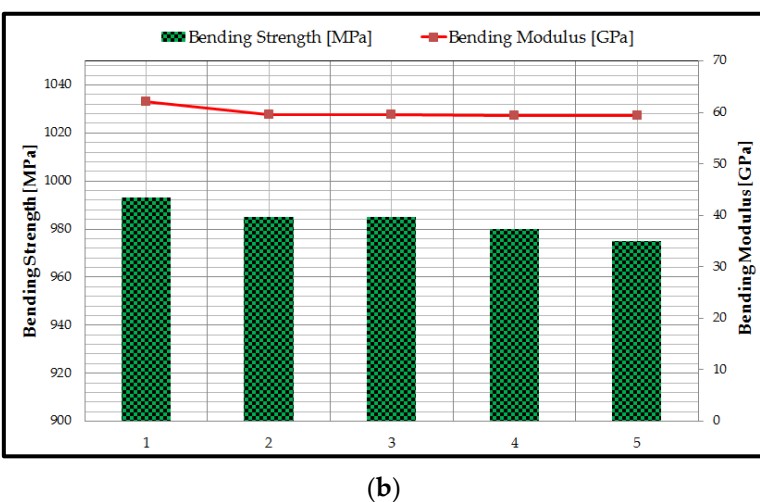

(**b**)

**Figure 6.** The results of three-point bending tests: (**a**) load–displacement curves of the specimens; (**b**) bending strength and bending modulus of specimens.

Using the software system of the WDW-150S testing machine, the main mechanical properties (bending strength and bending modulus) of the specimens manufactured by the SLS process from the A6 steel material were determined and graphically represented (Figure 6b). The bending performance of A6 steel specimens is higher or similar compared to the results obtained in other studies for different types of materials fabricated by additive manufacturing technologies [46,47] or traditional technologies [48–50]. The bending strength of tools' steel materials varies depending on the manufacturing technology and the heat treatment applied. In the case of steel alloys manufactured by additive technologies, the following values were obtained: the bending strength of steel 316 L manufactured by extrusion was 426.6 MPa [46], and for a deposition layer of 0.1 mm by the extrusion process of the 316 L material [47], it was 914.4 MPa (flatwise) and 792.2 MPa (edgewise). A6 tool steel and alloy steel manufactured by traditional technologies showed a bending strength in the range of 500 MPa and 990 MPa [41,48,49].

The main statistical indicators (mean, standard deviation, coefficient of variation) were calculated for the values of the bending strength and the bending modulus for the specimens manufactured using the SLS process from the A6 steel material (Table 5). For the data series above (the values of the bending strength and the bending modulus), the coefficient of variation was determined, in order to obtain an image of the homogeneity of the experimental data. From the results described in Table 5, it can be seen that the value of the maximum coefficient of variation is 1.8%. If the coefficient of variation (CV) is close to zero (CV < 30%), then the statistically processed data (the value of the coefficient of variation CV is between 0.6–1.8%) are homogeneous and the calculated mean is representative for these sets of values.

**Table 5.** The statistical indicators determined after the three-point bending tests of the A6 steel specimens.

| | Mean (m) | Standard Deviation (s) | Coefficient of Variation (CV)% |
|---|---|---|---|
| Bending strength [MPa] | 983.6 | 6.6 | 0.6 |
| Bending modulus [GPa] | 59.9 | 1.1 | 1.8 |

### 3.2. Tensile Behavior of Specimens Manufactured by the SLS Process

Static tensile tests were carried out on five specimens manufactured via the SLS process from A6 steel, until their breakage occurred. The load–displacement curves (Figure 7a) followed the same pattern for the specimens tested in bending, manufactured by the SLS process from A6 steel. From the graphic representations of the load–displacement curves, it can be seen that the maximum force (approximately 17 kN), at a displacement of 3.3 mm. Additionally, in the case of tensile tests, the behavior, from the point of view of the load–displacement curves, of the 5 specimens (Figure 7a), presents two main stages: linear growth, between the applied force and displacement, with some nonlinear behavior, towards the maximum curves and then a sudden decrease, at maximum force, at the moment of breaking the specimens.

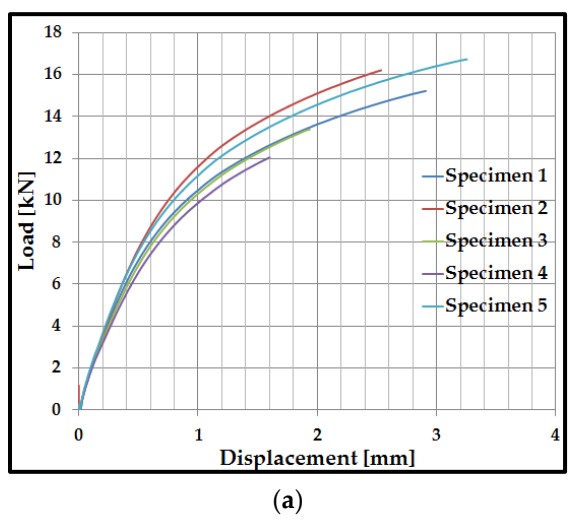
(a)

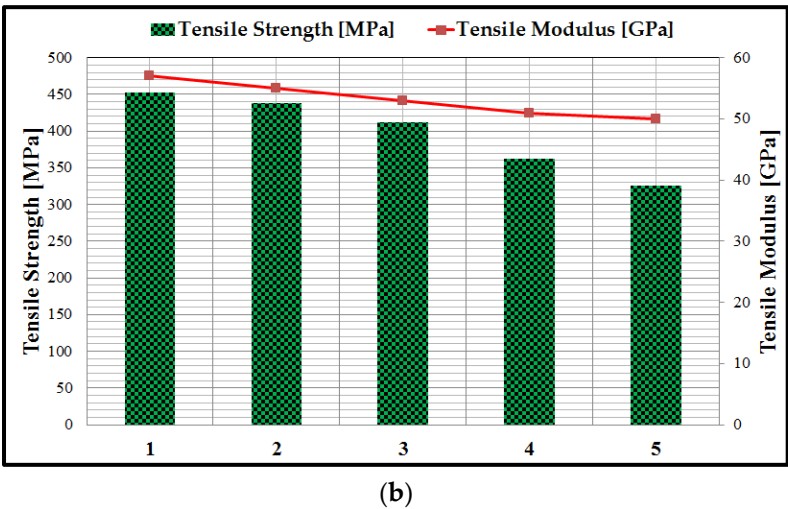
(b)

**Figure 7.** The results of tensile tests: (**a**) load–displacement curves of the specimens; (**b**) Tensile strength and tensile modulus of specimens.

This test method is used to investigate the mechanical performance (tensile strength, tensile modulus and behavior analysis resulting from the load–displacement curve) of specimens manufactured by the SLS process, from A6 steel. For the specimens manufactured by the SLS process from A6 steel, the software system of the machine allows the calculation of the following characteristics: tensile strength and tensile modulus of elasticity (Figure 7b). As can be seen from Figure 7b, the tensile strength varies between 326 MPa and 453 MPa, and the tensile modulus is in the range of 50 GPa and 57 GPa. This large variation can be caused by several factors, including positioning on the table or control over the infiltration process. The SLS additive manufacturing process is not 100% controllable and this is where these differences may arise. However, these differences are also found in other studies looking at the additive manufacturing process [50,51]. The tensile performance of A6 steel specimens is similar compared to the results obtained in other studies for different types of materials fabricated by additive manufacturing technologies [50,51] or traditional technologies [52,53]. The mechanical characteristics of the tools steels vary by the type of heat treatment applied and by the manufacturing process used to produce the specimens. Aluminum alloys manufactured by additive technologies presented the following perfor-

mances [50,51]: the tensile strength for 316 L manufactured by the SLM process indicated values between 510 MPa and 570 MPa, on the other hand, by the FFF process the tensile strength values were lower (between 300 MPa and 480 MPa). The tensile strength values for A6 steel tools [52] and aluminum alloys [53] are between 400 MPa and 520 MPa.

The values of the coefficient of variation (Table 6), for the obtained experimental data (tensile strength and tensile modulus), are relatively low. The maximum coefficient of variation for the tensile strength was 13.3%, and for the tensile modulus it was 5.4%. Thus, it can be appreciated that the experimental data are homogeneous and the mean is representative for the specimens manufactured via the SLS process using the A6 steel material.

**Table 6.** Statistical indicators determined from tensile tests of A6 steel specimens.

|  | Mean (m) | Standard Deviation (s) | Coefficient of Variation (CV)% |
|---|---|---|---|
| Tensile strength [MPa] | 398.6 | 53.1 | 13.3 |
| Tensile modulus [GPa] | 53.2 | 2.8 | 5.4 |

### 3.3. Microscopic Analysis of Specimens

The microscopic analysis of the specimens was carried out after the three sections established in Section 2.4. From the microscopic analysis of the three sections of the specimens, it can be seen that the specific process of obtaining metal parts with the help of the SLS process proceeded in good conditions, highlighting minor defects such as voids [54–56] of small sizes. The capillary infiltration process took place according to the technical conditions of the material and equipment used and at the optimal temperature, as proven by the micrographs analyzed (Figure 8), as well as the results obtained during the three-point bending and tensile testing of the specimens manufactured via the SLS process using A6 steel.

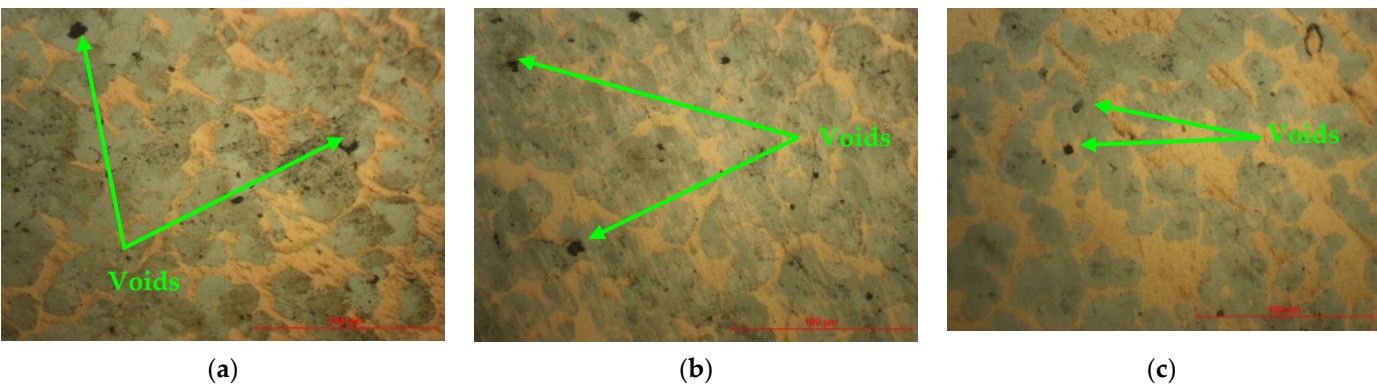

|  |  |  |
|---|---|---|
| (**a**) | (**b**) | (**c**) |

**Figure 8.** Microscopic analysis of specimens manufactured by the SLS process: (**a**) Section 1; (**b**) Section 2; (**c**) Section 3.

### 3.4. BLDC Motor Performance Analysis

The main motor characteristics were determined using the RCbenchmark 1585 test stand and the RCbenchmark GUI 1.2.0 software, namely: ESC signal [μs], torque [Nm], thrust [kgf], voltage [V], current [A], electric motor speed [RPM], electrical power [W], mechanical power [W], motor efficiency [%] and propeller mechanical efficiency [kgf/W]. To test the motor, a set of instructions was created to test automation, where the following modifications were made: the minimum and maximum values that the ESC signal can take; the number of steps and their duration; the number of repetition cycles.

What can be noted after testing the BLDC motor, with components manufactured by the SLS process, is that it shows high performance determined primarily by the most important result (thrust force). The maximum thrust force obtained during the tests was

4.68 kgf. This parameter was limited by the performance of the stand; thus, the motors functioned at approximately 75% of the thrust capacity. The results of the electric motor testing were analyzed by the outlier analysis methods (Grubbs test). Following the statistical processing, it can be concluded that no outliers were identified in the results of the electric motor testing. Using the Minitab 19 software system, the mean of the main performances of the BLDC motor was plotted. The performances of the BLDC motor were represented graphically, as follows: thrust—signal ESC (Figure 9a), torque—signal ESC (Figure 9b), current—signal ESC (Figure 9c), electrical power—signal ESC (Figure 9d), electrical speed and ESC signal (Figure 9e), mechanical power and ESC signal (Figure 9f), motor efficiency and ESC signal (Figure 9g), propeller mechanical efficiency and ESC signal (Figure 9h). It can be seen that all eight parameters analyzed in Figure 9 have a linear dependence that varies according to the ESC signal [57,58]. Additionally, the efficiency of the BLDC motor, defined as the ratio between the electric power and the mechanical power, shows an increase (up to about 90%) with the increase in the ESC signal (Figure 9f).

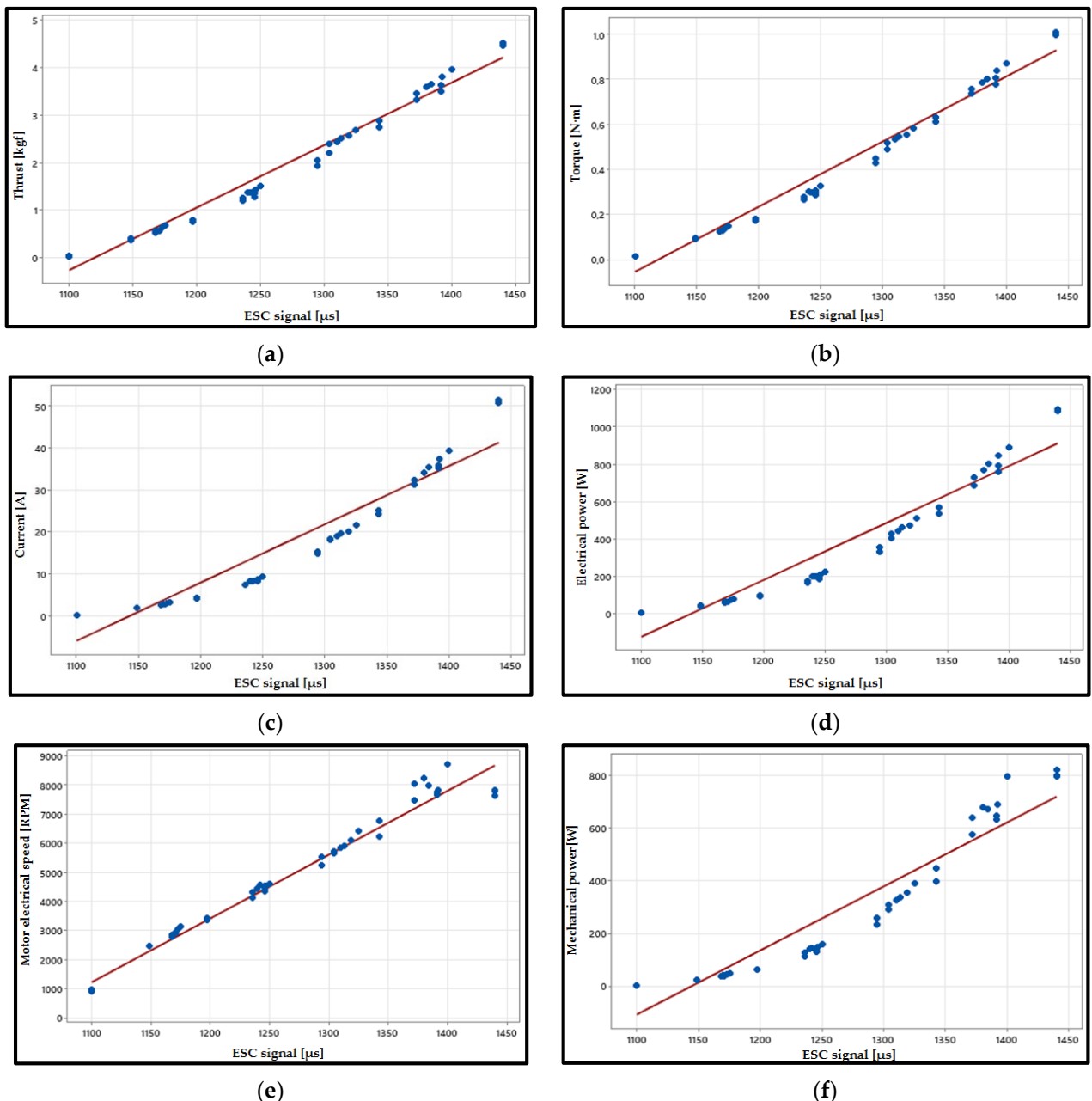

**Figure 9.** *Cont.*

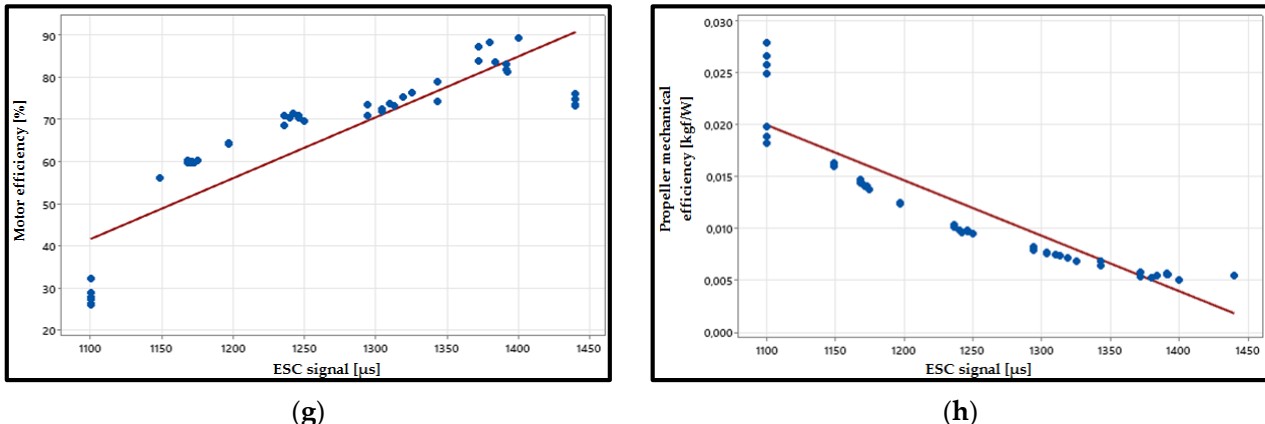

**(g)**                                                         **(h)**

**Figure 9.** BLDC motor test results: (**a**) Dependence between thrust and ESC signal; (**b**) dependence between torque and ESC signal; (**c**) dependence between current and ESC signal; (**d**) dependence between electrical power and ESC signal; (**e**) dependence between motor electrical speed and ESC signal; (**f**) dependence between motor mechanical power and ESC signal; (**g**) dependence between motor efficiency and ESC signal; (**h**) dependence between propeller mechanical efficiency and ESC signal.

The force generated by the BLDC motor, manufactured using the SLS process from A6 steel, at different rotation speeds is presented in Figure 10a. The maximum thrust generated by the BLDC motor was 4.68 kgf at 7800 RPM. Thrust force and torque are directly proportional to the square of the propeller rotation speed [57,58], as can be seen in Figure 10a,b.

The maximum torque generated by the propeller equipped BLDC motor is approximately 1 Nm at 8000 RPM, as indicated in Figure 10b. The efficiency of the BLDC motor (Figure 10c) is an important factor in determining whether the SLS motor and the ESC signal work well together. In general, the efficiency of the motor shows an increase with the increase in motor electrical speed, as can be seen in Figure 10c. The efficiency of the propeller is equal to the ratio between the output quantity—in the analyzed case, the thrust force—and the input quantity, which is mechanical power. As can be seen from Figure 10d, the propeller efficiency has a clear trend and relationship with the motor electrical speed parameter. Unlike thrust force, torque and motor efficiency, the propeller efficiency showed an inversely proportional dependence in relation to the motor electrical speed parameter.

The analyzed motors that were manufactured using classic technologies [59–61] have mass and dimensions approximately equal to those of the motor manufactured using additive technologies. Additionally, their performances (thrust force and motor efficiency) are similar to those of the motor manufactured via the SLS process. For these reasons, it can be concluded that there is similar behavior between the performance of the BLDC motor manufactured by the SLS process and the performance of the motors manufactured by classical technologies [59–61], conclusions which result from the comparative analysis carried out on the results of the testing of the two types of motors.

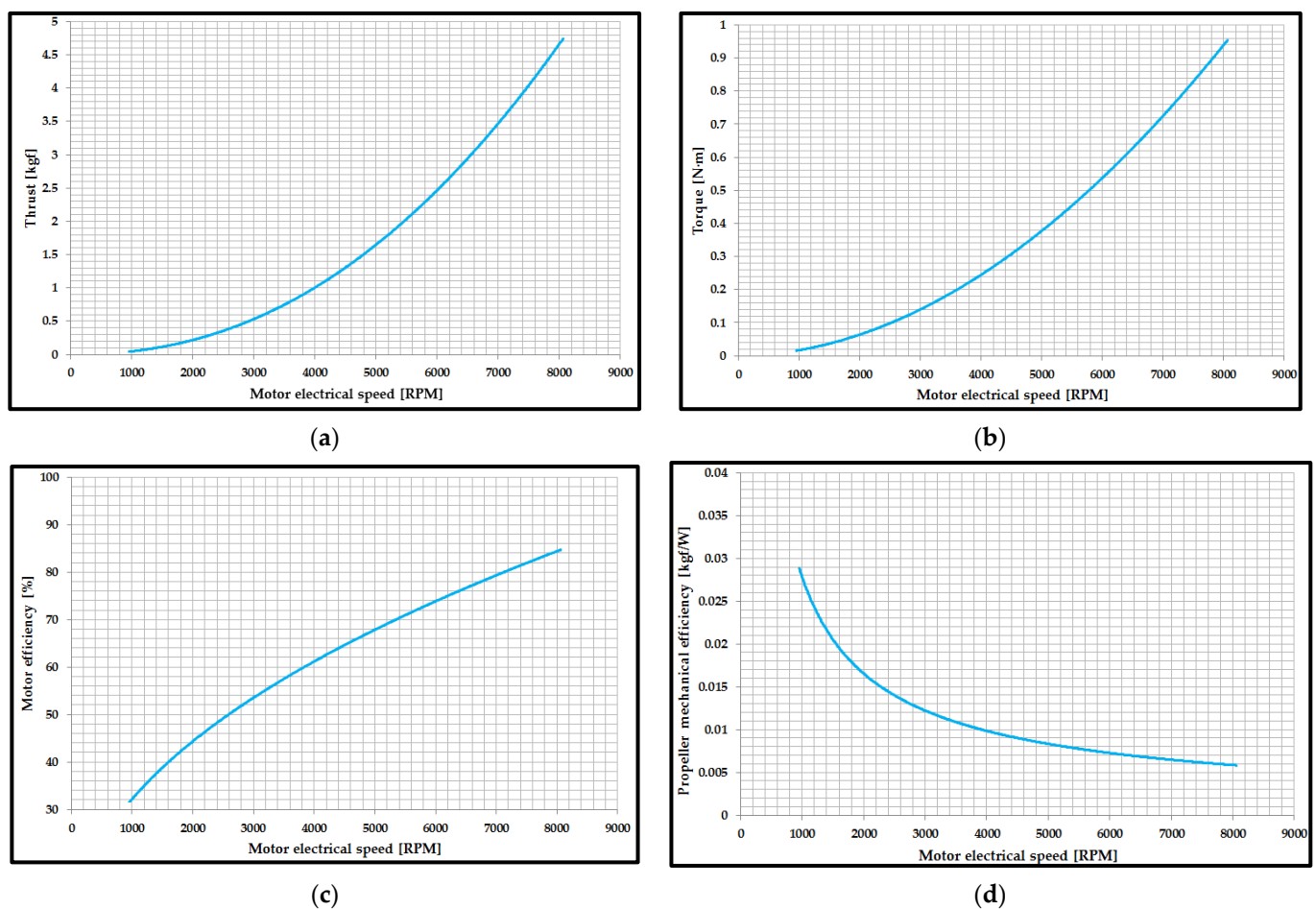

**Figure 10.** The performances of the BLDC motor manufactured via the SLS process: (**a**) thrust variation according to motor electrical speed; (**b**) variation of torque according to motor electrical speed; (**c**) motor efficiency variation according to motor electrical speed; (**d**) variation of propeller mechanical efficiency according to motor electrical speed.

## 4. Conclusions

Through metal additive manufacturing technology, more and more aerospace products can be made faster than ever before, which means that 3D metal powder prototyping is becoming a real and viable option for aircraft manufacturers. In the framework of this paper, the mechanical characteristics of the A6 steel material, manufactured via the SLS process, were studied, using three-point bending tests and tensile tests. The determination of the mechanical performances (three-point bending and tensile) of the specimens manufactured by the SLS process represented the first step in the manufacture of metal components for a BLDC motor. The mechanical characteristics of the A6 steel material, manufactured by the SLS process, had tensile strength (398.6 MPa) and bending strength (983.6 MPa) values close to those of the materials obtained by additive processes. The microscopic analysis of the material again revealed its good homogeneity after infiltration, with small defects (voids) specific to additive manufacturing processes.

After testing the BLDC motor, it can be highlighted that the main performances of the motor showed stable behavior and have a linear dependence with the increase in the ESC signal or motor electrical speed. Additionally, the most important parameter required for the use of this motor to power an UAV was determined, namely the thrust force having the value of 4.68 kgf. In conclusion, the SLS process can be successfully used to manufacture components from a BLDC motor prototype, and the advantages that this process confers are obvious: short time to build the components for the BLDC motor prototype, good

accuracy of the manufactured component (post processing of the parts required small-scale machining), and the manufactured BLDC motor components have high strength, which provides the possibility of manufacturing the BLDC motor components with complex geometries, followed by the testing and use of these motors to power an UAV in flight.

**Author Contributions:** Conceptualization, S.-M.Z., G.R.B. and L.-A.C.; methodology, S.-M.Z., M.A.P., L.-A.C. and V.M.S.; software, G.R.B. and C.L.; validation, S.-M.Z., G.R.B. and V.M.S.; investigation, S.-M.Z., M.A.P., G.R.B., V.M.S. and I.S.P.; writing—original draft preparation, S.-M.Z., G.R.B. and L.-A.C.; project administration, S.-M.Z. All authors have read and agreed to the published version of the manuscript.

**Funding:** This work was supported by a grant of the Ministry of Research, Innovation and Digitization, CNCS/CCCDI—UEFISCDI, project number PN-III-P2-2.1-PED-2019-0739, within PNCDI III.

**Data Availability Statement:** Not applicable.

**Acknowledgments:** We also acknowledge PRO-DD Structural Founds Project (POS-CCE, O.2.2.1., ID 123, SMIS 2637, ctr. No. 11/2009) for providing the infrastructure used in this work.

**Conflicts of Interest:** The authors declare no conflict of interest.

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
