# Peer review of "Design and Testing of Brushless DC Motor Components of A6 Steel Additively Manufactured by Selective Laser Sintering"

_aerospace, doi:10.3390/aerospace10010060_

Round 1

Reviewer 1 Report

In general, the quality of writing is poor. There is excessive use of commas in the text. The phrases are poorly constructed.

There is no comparison between motors manufactured by the traditional manufacturing process and AM-manufactured motors. There is no indication if the general performance of the motor is improved or not.

The results of the mechanical tests SLS manufactured with A6 steel parts are represented, however, there is no comparison between SLS manufactured parts with samples made by traditional manufacturing with A6 steel. 

There is no clear explication of why the authors have selected the 3Point bending test and tensile test for determination of the mechanical strength. As in this case there are more important performance indexes such as mass or vibration.

There is CAD images of the tensile and bending specimen, I don't understand what is the added value when such images are added to the texts.

The detailed comments are added to the pdf file attached bellow.

Reviewer 2 Report

From the overall structure of the paper, this paper is a good application of additive manufacturing technology. The overall idea of the paper is clear and the writing is standardized, therefore, it is agreed to be accepted.

Introduction: There are too many paragraphs in the Introduction section. It is suggested that the structure should be three paragraphs: "background + application of additive manufacturing technology + content of this study".

Figure 10: The code presentation in Figure 10a appears too arbitrary and is best presented in a pseudo-code manner.

Conclusions: The conclusion section is overloaded with text and is recommended to be streamlined.

Reviewer 3 Report

COMMENTS ON AEROSPACE-1949720

TITLE: Design and Testing of Brushless DC Motor Components of A6 Steel Additively Manufactured by Selective Laser Sintering

In this paper, the authors explore the mechanical properties of A6 steel material, produced by the SLS process, by means of three-point bending and tensile tests. This represented a first step in the production of metal components for a BLDC motor.

The reviewer believes that the paper is technically accurate and offers interesting informations. Hence the paper is recommended for publication on condition that the changes suggested by the reviewer are complied with.

1)              Abstract: please replace the sentence in line 11 with the following one “This paper presents the design, fabrication by the selective laser sintering process, and testing of the mechanical performance by performing three-point bending and tensile tests on A6 steel specimens”;

2)              Section 2: add the geometric information described in lines 118-121 also in Figure 2, please.

3)              Subsection 2.3: “The other components of the motor (stator with winding, shaft) were manufactured using classical manufacturing technologies”. Which ones?

4)              Is this infiltration process with bronze necessary only for sintering this material? Can the authors include some references to clarify this? I have never heard of the need for this infiltration process to print metals with SLS technology.

5)              Subsection 2.6: please replace “Kgf” with “kgf”.

6)              What standards were adopted to perform the mechanical tests?

7)              The conclusions are lacking of a quantitative description of the achievements of this research work.

8)              The reviewer suggests adding reference on application examples of additive manufacturing. Here are some suggestions:

·       Acanfora, Valerio, Mauro Zarrelli, and Aniello Riccio. "Experimental and numerical assessment of the impact behaviour of a composite sandwich panel with a polymeric honeycomb core." International Journal of Impact Engineering 171 (2023): 104392.

Reviewer 4 Report

The manuscript proposes the feasibility of designing, manufacturing by the selective laser sintering process, and testing of components from a brushless motor, made from A6 steel material. Moreover, the mechanical performance in three-point bending and traction of A6 steel specimens was demonstrated.

The work is interesting and well-structured. It should be considered for publication in this journal after a review taking into account the following points:

·         There are some typos in the text (i.e. line 37: ”However, an accurate estimate of the performance of BLDC motors…” or line 98: ”By using current standards [38,39] that applie to metal specimens manufactured by additive processes…”) and some sentences could be expressed more clearly (i.e. the sentence from line 39 to line 51). Please review the document.

·         Image 10a is unclear. Replace it with a better-quality image or consider deleting it.

·         Sections 3.1 and 3.2 are very similar. Consider merging them into one section on the mechanical characterization of specimens to improve the readability of the work.

·         Please add in sections 3.1 and 3.2 literature references on the mechanical characteristics of A6 steel specimens produced by AM techniques.

·         In section 3.3 on the microscopic analysis of specimens it would be appropriate to include more references concerning the mechanical characterization and typical defect of the component produced by AM processes. Some references to consider are:

·         https://doi.org/10.1007/s11665-021-05919-6

·         https://doi.org/10.3390/ma13112658

·         https://doi.org/10.1016/j.matdes.2020.108708

·         In section 3.4 the performance of the engine made with additive technology was evaluated. It would be appropriate to include a comparison with the performance of similar versions of the engine produced by traditional techniques, to highlight the effectiveness of this design.

Round 2

Reviewer 1 Report

Regarding the answers in cover letter, Please apply these minor modifications ( nombers are the number of ther questions in the cover letter.

B.

Engine is for the combustion motors. And could not be used in this case. Change the word.

c. 

For the sake of comparison please add the A6 tool properties ( or the properties of most used steel for the motor) in the text and the figures. So the reader could conclude compare the properties of Laserform A6 steel with the traditional material used.

D.

I am not still convinced for the answer. Tensile test could be acceptable for having a general knowledge on the properties. However, in case of  bending I am not sure.

 5.

Add this phrase to the text please.

34. 

Please add the reason of the deviation of samples in the text.

Reviewer 3 Report

It is ok

Author Response

Thank you for the review.

Reviewer 4 Report

The manuscript has been modified in accordance with the request. However, there are a few more changes to make before publishing:

·         In section 3.2 the results of the tensile tests on the produced samples were shown. However, a very wide range of tensile strength is recorded (326 MPa and 453 MPa). Please provide an explanation for such a high variability and indicate if such variability was considered in the design of the components.

·         In section 3.4 the performance of the engine produced by SLS and the commercially available models were compared. However, references 60, 61 and 62 refer to internet retailing sites (Amazon and Hobbyking). These are unsuitable for a scientific article. Please modify by inserting references to the manufacturer's website/data sheet or to scientific works in the literature.
